# Elucidation of Inverse Agonist Activity of Bilastine

**DOI:** 10.3390/pharmaceutics12060525

**Published:** 2020-06-08

**Authors:** Hiroyuki Mizuguchi, Tomoharu Wakugawa, Hisato Sadakata, Seiichiro Kamimura, Mai Takemoto, Tomomi Nakagawa, Masami Yabumoto, Yoshiaki Kitamura, Noriaki Takeda, Hiroyuki Fukui

**Affiliations:** 1Laboratory of Pharmacology Faculty of Pharmacy Osaka Ohtani University, Osaka 584-8540, Japan; u4115067@osaka-ohtani.ac.jp (M.T.); u4115080@osaka-ohtani.ac.jp (T.N.); 2Department of Molecular Pharmacology, Institute of Biomedical Sciences, Tokushima University Graduate School, Tokushima 770-8505, Japan; c401741003@tokushima-u.ac.jp; 3TIMELAPS VISION INC., Saitama 353-0004, Japan; sadakata@timelapsevision.com; 4Department of Otolaryngology, Institute of Biomedical Sciences, Tokushima University Graduate School, Tokushima 770-8505, Japan; c201756004@tokushima-u.ac.jp (S.K.); ykitamura@tokushima-u.ac.jp (Y.K.); takeda@tokushima-u.ac.jp (N.T.); hfukui@tokushima-u.ac.jp (H.F.); 5Medical Corporation Kinshukai, Osaka 558-0011, Japan; yabumoto.masami@kinshukai.or.jp

**Keywords:** H_1_-antihistamines, histamine H_1_ receptor gene expression, inositol phosphates accumulation, inverse agonist, time-laps Ca^2+^ imaging

## Abstract

H_1_-antihistamines antagonize histamine and prevent it from binding to the histamine H_1_ receptor (H1R). Some of them also act as inverse agonists, which are more potent than pure antagonists because they suppress the constitutive H1R activity. Bilastine is a non-sedative antihistamine which is one of the most satisfy the requirements for oral antihistamines. However, there is no information to show the inverse agonist activity of bilastine including inositol phosphates accumulation, and its inverse agonist activity is yet to be elucidated. Here we evaluated whether bilastine has inverse agonist activity or not. Intracellular calcium concentration was measured using Fluo-8. Inositol phosphates accumulation was assayed using [^3^H]myo-inositol. The H1R mRNA level was measured using real-time RT-PCR. At rest, Ca^2+^ oscillation was observed, indicating that H1R has intrinsic activity. Bilastine attenuated this fluorescence oscillation. Bilastine suppressed the increase in IPs formation in a dose-dependent manner and it was about 80% of the control level at the dose of 3 μM. Bilastine also suppressed histamine-induced increase in IPs formation to the control level. Furthermore, bilastine suppressed basal H1R gene expression in a dose-dependent manner. Data suggest that bilastine is an inverse agonist. Preseasonal prophylactic administration with bilastine could down-regulate basal H1R gene expression in the nasal mucosa and ameliorate the nasal symptoms during the peak pollen period.

## 1. Introduction

Histamine is an important chemical mediator causing symptoms of pollinosis [1]. Its action occurs mainly through the activation of histamine H_1_ receptor (H1R). Previously, we demonstrated that H1R gene expression is correlated with the severity of nasal symptoms in toluene-2,4-diisocyanate (TDI)-sensitized rats and patients with pollinosis [2,3,4]. We also showed that protein kinase Cδ (PKCδ) signaling was involved in H1R gene expression, and that suppression of the H1R gene up-regulation alleviated these nasal symptoms in TDI-sensitized rats [5,6,7], suggesting that keeping the H1R gene expression level low is effective for improving nasal symptoms. H1R is in an equilibrium state between an active form and an inactive form, and a constant level of signal always operates even without histamine stimulation [8]. Although it is well-known that H_1_-antihistamines antagonize histamine and prevent it from binding to the H1R and are widely used as the first-line medicine for nasal symptoms of pollinosis, some H_1_-antihistamines also act as inverse agonists that bind to and stabilize the inactive form of H1R, and down-regulate constitutive receptor activity. Therefore, H_1_-antihistamines with inverse agonist activity are more potent than neutral antagonists, as they suppress this intrinsic signal in addition to the H_1_-antihistamine effect. Recently, we have demonstrated that HeLa cells expressing H1R endogenously are useful to assess inverse activity of H_1_-antihistamines [9,10,11].

Bilastine (Figure 1A) is a recently launched new H_1_-antihistamine, which has high affinity for H1R. It has been demonstrated to have a potent anti-allergic activity [12,13]. Bilastine is classified as the non-sedative H_1_-antihistamines because its brain H1R occupancy (H1RO), which has been used for as the index of sedative potential of H_1_-antihistamines, is nearly 0% [14]. Conclusively, bilastine is described as the most satisfy the requirements for oral H_1_-antihistamines in Allergic Rhinitis and its Impact on Asthma (ARIA) guidelines [15]. However, its inverse agonist activity is yet to be elucidated.

In the present study, we sought to elucidate the inverse agonist activity of bilastine. To assess the inverse activity of bilastine, the following three methods were used; (1) time-laps Ca^2+^ imaging; (2) inositol phosphates (IPs) accumulation; and (3) H1R gene expression in HeLa cells expressing H1R endogenously.

## 2. Materials and Methods

### 2.1. Chemicals

Bilastine was supplied by Faes Farma (Leioa, Spain). Fexofenadine hydrochloride and olopatadine hydrochloride were from Cayman Chemical (Ann Arbor, MI, USA). Oxatomide was from Toronto Research Chemicals (North York, ON Canada). Levocetiridine dihydrochloride was from Combi-Blocks (San Diego, CA, USA). Bepotastine beslilate was from ChemScence (Monmouth Junction, NJ, USA). Rupatadine fumarate was from MedChemexpress (Monmouth Junction, NJ, USA). Desloratadine was from Sigma-Aldrich (St, Louis, MO, USA). *myo*-[^3^H]-inositol (37kBq/well) and Insta-Gel Plus scintillation cocktail were from PerkinElmer (Waltham, MA, USA). AG 1-X8 (200–400 mesh formate form) was from Bio-Rad (Richmond, CA, USA). The predeveloped TaqMan Assay Reagent of human glyceraldehyde-3-phosphate dehydrogenase (GAPDH) was from Applied Biosystems (Foster City, CA, USA). Minimal essential medium-α (MEM-α) was from Invitrogen (Carlsbad, CA, USA). RNAiso Plus was purchased from Takara Bio (Kyoto, Japan). Rottlerin was purchased from Merk Millipore (Billerica, MA, USA). All other chemicals were of analytical grade.

### 2.2. Intracellular Ca^2+^ Measurement Ca^2+^ Imaging

Intracellular Ca^2+^ concentration was measured using time-lapse microscopy as described previously [11]. HeLa cells (1.5 × 10^5^ cells) were cultured in 12-mm glass bottom dishes and incubated in in MEM-α for 24 h. The cells were treated with H_1_-antihistamines (or DMSO as a control) in MEM-α containing 10 μM Fluo-8, 0.04% pluronic F-127, and 1 nM probenecid for 3 h at 37 °C under 5% CO_2_. Then, the cell-associated Fluo-8 fluorescence intensity was monitored for 20 min at every 22 s in the confocal microscope (LSM-700, Carl Zweiss, Oberkochen, Germany). Fluorescence intensity was visualized by 256 color images and scored from 0 to 255 using the apparatus-attached software. Oscillation was defined as the fluorescence intensity increased by 5 or more during the 22-s interval. The frequency (oscillation frequency) and the value of increase (oscillation intensity) were also calculated.

### 2.3. Measurement of [^3^H]-Inositol Phosphates Formation

HeLa cells seeded in 24-well plates (2 × 10^5^ cells/well) were incubated in MEM-α for 24 h. They were labeled with *myo*-[^3^H]-inositol (37 kBq/well) in medium 199 (Invitrogen) supplemented with 8% fetal calf serum (Invitrogen) by another 24-h incubation. The cells were then washed twice with 0.5 mL of HEPES-buffered saline solution (HBS, 125 mM NaCl, 4.7 mM KC1, 1.2 mM MgCl_2_, 1.2 mM KH_2_PO_4_,15 mM NaHCO_3_, 11 mM glucose, and 15 mM HEPES, pH 7.4) containing 10 mM LiCl. H_1_-antihistamines with or without 100 μM histamine was added to the HBS with LiCl and incubated for 2.5 h. After the incubation, the cells were washed with 0.5 mL of HBS+LiCl, and the reaction was terminated by the quick aspiration of the incubation medium and the addition of 0.5 mL of 5% (*w/v*) trichloroacetic acid (TCA) solution to the cells. The cell suspensions were transferred to new tubes and neutralized with 0.1 mL of 0.2 M Tris. After centrifugation at 17,400g for 15 min at 4 °C, the resulting supernatant was applied on a AG 1-X8 (200–400 mesh, 0.5 mL bed, BioRad,(Richmond, CA, USA) pre-equilibrated with 5 mL of 3 M ammonium formate, and 0.1 M formic acid. The column was washed twice with 5 mL each of 5 mM myoinositol. Then, IPs were eluted with 5 mL of 1 M ammonium formate, and 0.1 M formic acid. The resulting precipitate was dissolved in 0.2 mL of 1% SDS/0.2 M NaOH and designated as total count. Radioactivity in the eluates was determined by a scintillation counter using Insta-Gel Plus. IPs accumulated was calculated as follows;
IPs accumulated = radioactivity in supernatant/radioactivity in total count.

### 2.4. Real-Time Quantitative RT-PCR

HeLa cells were cultured at 37 °C under a humidified 5% CO_2_, 95% air atmosphere in MEM-α containing 8% fetal calf serum (Sigma-Aldrich, St, Louis, MO, USA), 10,000 Units/mL penicillin G sodium (Sigma-Aldrich), and 10 mg/mL streptomycin (Sigma-Aldrich). HeLa cells cultured to 80% confluency in 6-well dishes were serum-starved for 24 h and then treated with H_1_-antihistamines for 4 h. Then, the cells were harvested with 700 μL of RNAiso Plus, mixed with 140 μL of chloroform, and centrifuged at 17,400 g for 15 min at 4 °C. The aqueous phase was collected, and RNA was precipitated by the addition of isopropyl alcohol. After centrifugation at 17,400 g for 15 min at 4 °C, the resulting RNA pellet was washed with ice-cold 70% ethanol. Total RNA was resuspended in 10 μL of diethylpyrocarbonate-treated water. The RNA samples (2 μg) were reverse transcribed to cDNA using a PrimeScript RT Reagent Kit (Takara Bio). TaqMan primers and the probe were designed using Primer Express (Applied Biosystems, Foster City, CA, USA). Real-time PCR was conducted using a GeneAmp 7300 sequence detection system (Applied Biosystems). The sequences of the primers and TaqMan probe were as follows: forward primer, 5′-CAGAGGATCAGATGTTAGGTGATAGC-3′; reverse primer, 5′-AGCGGAGCCTCTTCCAAGTAA-3′; TaqMan probe, FAM-CTTCTCTCGAACGGACTCAGATACCACC-TAMRA. To standardize the starting material, TaqMan Assay Reagent of human glyceraldehyde-3-phosphate dehydrogenase (GAPDH) (Applied Biosystems) was used, and data were expressed as the ratio of H1R mRNA to GAPDH mRNA.

### 2.5. Statistical Analysis

The results are shown as means ± SEM. Statistical analyses were performed using unpaired t-tests or ANOVA with Dunnett’s multiple comparison test using the GraphPad Prism software 6 (GraphPad Software, Inc., La Jolla, CA, USA). *p* < 0.05 was considered statistically significant.

## 3. Results

### 3.1. Effect of Bilastine on Intracellular Ca^2+^ Elevation by Constitutive and Agonist-Induced H1R Activation

Intracellular Ca^2+^ oscillation was observed in HeLa cells at rest (Figure 1B). The oscillation frequency and the intensity were 9.7 ± 0.8 times/20 min, and 11.3 ± 0.5, respectively (*n* = 12, Figure 1C). Stimulation of the cells with 1 mM histamine immediately increased in the fluorescence intensity (7.5 ± 2.1 for vehicle (DMSO) vs. 66.5 ± 13.2, during 2 min after histamine stimulation), confirming the expression of H1R (Figure 1D).

Pretreatment with bilastine or fexofenadine attenuated fluorescence oscillation observed in HeLa cells at rest (Figure 2A). Oscillation frequency was decreased from 10.1 ± 0.49 to 7.6 ± 0.55 (bilastine) and 8.6 ± 0.55 (fexofenadine, Figure 2B). Bilastine significantly decreased the oscillation intensity (10.5 ± 0.48 for vehicle vs. 9.2 ± 0.48, Figure 2C). Fexofenadine tended to decrease the oscillation intensity (10.5 ± 0.48 vs. 9.3 ± 0.44). Fluorescence intensity was also significantly decreased (104.4 ± 4.6 for vehicle vs. 65.9 ± 2.2 for bilastine or 68.9 ± 2.5 for fexofenadine, Figure 2D). On the contrary, treatment with rupatadine did not show any significant difference in oscillation frequency (6.4 ± 0.58 for vehicle vs. 6.5 ± 0.44), oscillation intensity (9.5 ± 0.62 for vehicle vs. 11.0 ± 0.73), or fluorescence intensity (79.6 ± 8.4 for vehicle vs. 79.1 ± 5.3, Figure 3).

### 3.2. Effect of Bilastine on IPs Accumulation by Constitutive and Agonist-Induced H1R Activation

Pretreatment of HeLa cells with bilastine (0.3 to 3 μM) for 2.5 h suppressed IPs formation in a dose-dependent manner and it was about 80% of the control level at the dose of 3 μM (Figure 4A). Stimulation with 100 μM of histamine, IPs formation was increased about twice as compared to control (Figure 4B). Bilastine significantly suppressed histamine-induced increase in IPs formation to the control level (Figure 4B).

### 3.3. Effect of Bilastine on Constitutive H1R Gene Expression

Pretreatment with bilastine (0.3 to 3 μM) for 4 h suppressed basal H1R gene expression in HeLa cells in a dose-dependent manner (Figure 5A). Fexofenadine also suppressed basal H1R gene expression (Figure 5B). Levocetirizine and bepotastine tended to suppress basal H1R gene expression, while olopatadine and oxatomide did not show any significant change of the basal H1R mRNA level (Figure 5B). No suppression of basal H1R gene expression was observed by the pretreatment of rupatadine or desloratadine (Figure 6)

## 4. Discussion

In the present study, we elucidated the inverse agonist activity of bilastine. We used HeLa cells that express H1R endogenously in this study. Stimulation with histamine induced intracellular Ca^2+^ elevation followed by IP_3_ and diacylglycerol formation, then caused the increase in H1R gene expression through the activation of H1R [16]. Therefore, HeLa cells are useful to evaluate inverse agonist activity of H_1_-antihistamines. Using HeLa cell system, we have reported that carebastine, mepyramine, fexofenadine, cetirizine, chlorpheniramine, diphenhydramine, and epinastine were inverse agonists and oxatomide, loratadine, and olopatadine were neutral antagonists [9,10,11]. According to our previous data, we selected fexofenadine and levocetirizine as controls of inverse agonists, and olopatadine and oxatomide as controls of neutral antagonists. We also selected rupatadine because it is partially metabolized to desloratadine that is an active form of loratadine. In addition, we selected bepotastine because it is one of the most prescribed H_1_-antihistamines in Japan.

Firstly, we investigated the effect of bilastine on Ca^2+^ mobilization. H1R is coupled with Gq and its activation increases intracellular Ca^2+^ concentration followed by IP_3_ and diacylglycerol production. At rest, HeLa cells showed Ca^2+^ oscillation derived from the constitutive activity of H1R. Bilastine reduced this constitutive activity. Fexofenadine, known as another inverse agonist also attenuated the Ca^2+^ oscillation. However, similar to olopatadine [11], rupatadine did not show any effect on basal H1R activity, suggesting that rupatadine is a neutral antagonist. Bilastine also suppressed basal IPs formation by constitutive activation of phospholipase C in response to H1R constitutive activity. Furthermore, bilastine suppressed basal H1R gene expression in HeLa cells. Fexofenadine also suppressed H1R gene expression and levocetirizine and bepotastine tended to suppress H1R gene expression, suggesting that these H_1_-antihistamines have inverse agonist activity. It was reported that desloratadine, an active metabolite of rupatadine, was an inverse agonist [17]. However, our data from time-laps Ca^2+^ imaging study demonstrated that rupatadine was a neutral antagonist. Therefore, we investigated the effect of rupatadine and desloratadine on basal H1R gene expression. Data showed that rupatadine or desloratadine did not show any suppression on basal H1R gene expression, suggesting these antihistamines were neutral antagonists.

Crystallographic studies hypothesized that Trp^428^ in helix VI works as a key molecular switch in the process of receptor activation [18]. It was also reported that Ile^420^ was an important residue for inverse agonist activity, which interacts with Asn^464^ and restrains the side chain of Asn^464^ toward transmembrane domain 6 in the inactive state of the receptor [19]. It was demonstrated that doxepin, classified as an inverse agonist [20], made extensive hydrophobic interaction with Trp^428^ ring [18]. Mutation of Ile^420^ to Arg or Lys also resulted in high constitutive activity [19]. Docking model studies of bilastine-H1R complex revealed that bilastine and fexofenadine bind to H1R with similar manner [14]. Similar to fexofenadine, bilastine is a carboxy-group-containing zwitterionic H_1_-antihistamine with big molecular size [14]. So, it is likely that bulkiness of bilastine could block both Ile^420^-Asn^464^ interaction that participates in histamine-induced conformational change and Trp^428^ switch, and lock H1R in the inactive form. Our data indicates that olopatadine is a neutral antagonist. However, crystallographic studies showed that binding mode of olopatadine to H1R closely resembled that of doxepin [18]. Closer examination of structure of olopatadine-H1R complex suggests that carboxyl group of olopatadine extends out of the ligand binding pocket and cannot block Ile^420^-Asn^464^ interaction. We think this is the reason why olopatadine does not have inverse agonist activity.

Although structure-activity relationships (SAR) of H_1_-antihistamines have been well investigated, little information is available concerning SAR for their inverse agonist activity. There are six main chemical groups of H_1_-antihistamines: alkylamines, ethanolamines, ethylenediamines piperazines, piperidines, and phenothiazines [14]. Bilastine belongs to piperidine derivatives although it is not structurally derived from any other currently available antihistamines [21]. Fexofenadine and bepotastine are also piperidine derivatives, thus, piperidine moiety may contribute inverse agonist activities. However, desloratadine, another piperidine derivative, did not show inverse activity, suggesting the necessity of additional moiety for inverse agonist activity. We showed that levocetirizine, which is a piperazine derivative, is an inverse agonist, while, oxatomide, which belongs to the same group as levocetirizine, is a neutral antagonist. In addition, we reported that mepyramine, diphenhydramine, and chlorpheniramine (belongs to ethylamines, ethanolamines, and alkylamines, respectively) are also inverse agonists [9,10,11]. Thus, it is difficult to explain which structures contribute to inverse agonist activity of H_1_-antihistamines. Further studies should be addressed to SAR of inverse agonist activity of H_1_-antihistamines to understand the molecular mechanism how inverse agonists stabilize the inactive form of H1R. It was reported that some indolylpiperidines show H_1_-antagonistic activity [22]. Examination of their inverse agonist activity may provide some useful insights.

H_1_-antihistamines are widely used as the first-line medicine for pollinosis. Among them, inverse agonists are thought to be more potent than neutral antagonists, as they suppress this intrinsic histamine signaling in addition to the H_1_-antihistamine effect. According to the H1RO, it is considered that bilastine as well as fexofenadine are distinguished from other second-generation H_1_-antihistamines as non-brain-penetrating H_1_-antihistamines and are the most satisfy the requirements for oral H_1_-antihistamines in ARIA guidelines [15]. Bilastine has high affinity for H1R with the Ki value of 8.7 nM, thus, 20 mg once daily is therapeutically used for pollinosis [23,24]. In addition, we showed that bilastine is an inverse agonist. Together with other pharmacological characteristics of bilastine [12], these findings suggest that bilastine may be one of the best H_1_-antihistamines for pollinosis.

In conclusion, our data elucidated that bilastine has an inverse agonist activity. H1R gene is a diseases-sensitive gene for pollinosis and suppression of up-regulated H1R gene expression improves nasal symptoms. Therefore, by taking bilastine, which is an inverse agonist, before the pollinosis season, the H1R gene expression level can be kept low, so the nasal symptom ameliorating effect can be expected even in the pollinosis season.

## Figures and Tables

**Figure 1 pharmaceutics-12-00525-f001:**
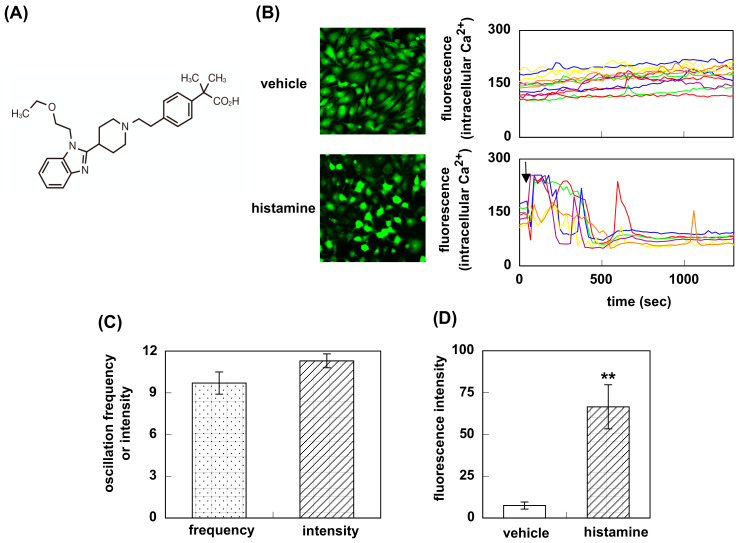
Intracellular Ca^2+^ elevation in HeLa cells. (**A**) Structure of bilastine. (**B**) HeLa cells were stimulated with 1 mM histamine (or vehicle) and intracellular Ca^2+^ was monitored for 20 min at 22 sec intervals. In (**C**), oscillation frequency and oscillation intensity for 20 min were counted from the data from vehicle-treated cells. Data are expressed as means ± SEM (*n* = 12). In (**D**), fluorescence intensity during 2 min after histamine (or vehicle) stimulation was calculated. Arrow shows the time for histamine stimulation. Data are expressed as means ± SEM (*n* = 6 or 12). **, *p* < 0.01 vs. vehicle.

**Figure 2 pharmaceutics-12-00525-f002:**
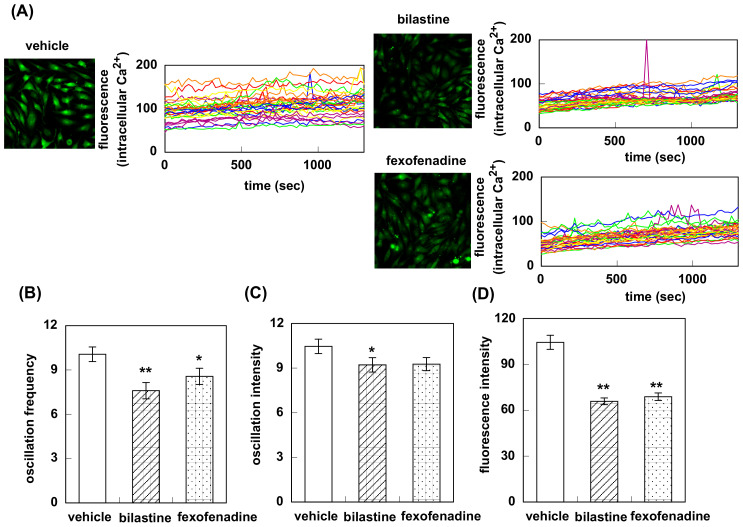
Effect of bilastine and fexofenadine on intracellular Ca^2+^ elevation in response to the constitutive activation of histamine H_1_ receptor. (**A**) HeLa cells were treated with 10 μM of bilastine or fexofenadine for 2 h at 37 °C and intracellular Ca^2+^ was monitored for 20 min at 22 sec intervals. In (**B**) to (**C**), oscillation frequency (**B**), oscillation intensity (**C**), and fluorescence intensity (**D**) for 20 min were counted. Data are expressed as means ± SEM (*n* = 30). **, *p* < 0.01; *, *p* < 0.05 vs. vehicle.

**Figure 3 pharmaceutics-12-00525-f003:**
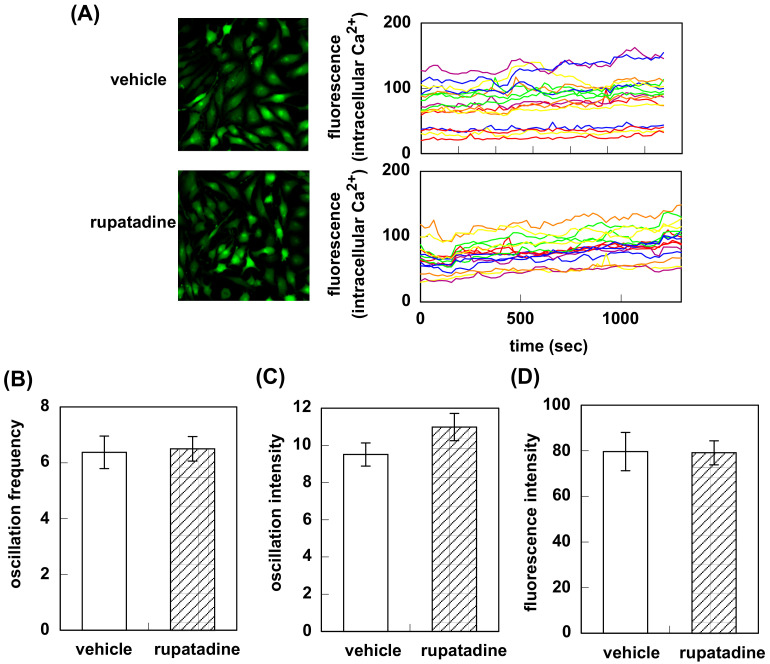
Effect of rupatadine on intracellular Ca^2+^ elevation in response to the constitutive activation of histamine H_1_ receptor. (**A**) HeLa cells were treated with 10 μM of rupatadine for 2 h at 37 °C and intracellular Ca^2+^ was monitored for 20 min at 22 sec intervals. In (**B**) to (**C**), oscillation frequency (**B**), oscillation intensity (**C**), and fluorescence intensity (**D**) for 20 min were counted. Data are expressed as means ± SEM (*n* = 16).

**Figure 4 pharmaceutics-12-00525-f004:**
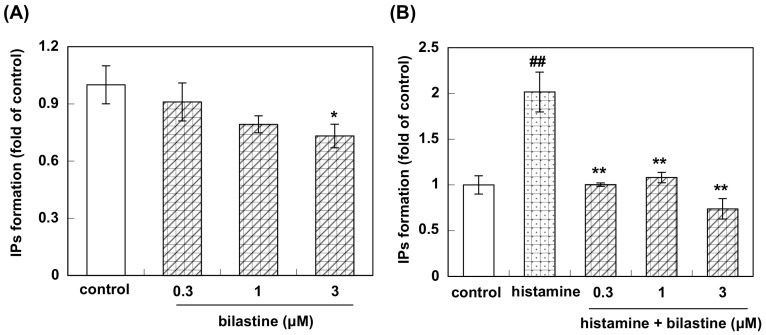
Effect of bilastine on accumulation of IPs in HeLa cells. (**A**) Dose–response study; HeLa cells were incubated with or without bilastine (0.3 μM to 3 μM) for 2.5 h. Then accumulation of IPs was determined as described in the Material and methods. Data are expressed as means ± SEM (*n* = 8 for control, *n* = 9 for 3 μM bilastine, and *n* = 6 for 0.3 and 1 μM bilastine). * *p* < 0.05 vs. control. In (**B**), HeLa cells were stimulated with 100 μM histamine for 2.5 h with or without bilastine (0.3 μM to 3 μM). Data are expressed as means ± SEM (*n* = 8 for control, *n* = 9 for 3 μM bilastine, *n* = 6 for 1 μM bilastine, and *n* = 5 for 0.3 μM bilastine). ##, *p* < 0.01 vs. control; **, *p* < 0.01; *, *p* < 0.05 vs. histamine.

**Figure 5 pharmaceutics-12-00525-f005:**
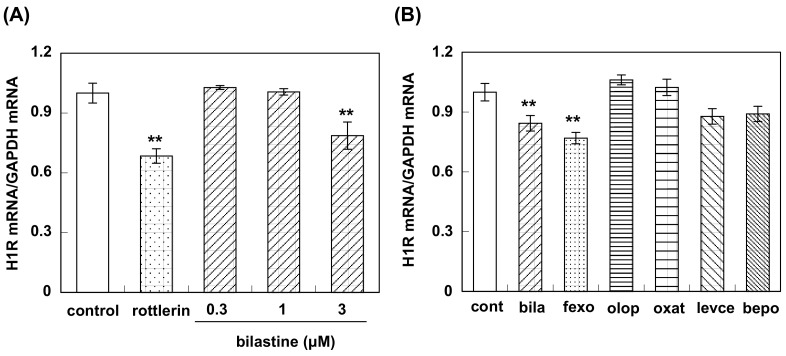
Effect of antihistamines on H1R gene expression in HeLa cells. (**A**) Dose–response study; HeLa cells were incubated with or without 5 μM rottlerin or bilastine (0.3 μM to 3 μM) for 4 h. Then the cells were harvested and H1R mRNA was determined by real-time RT-PCR. Data are expressed as means ± SEM (*n* = 8 for control and bilastine (0.3 and 3 μM); *n* = 4 for bilastine (1 μM) and rottlerin). **, *p* < 0.01 vs. control. In (**B**), HeLa cells were treated with or without (represented as control) 3 μM H_1_-antihistamines for 4 h. cont, control; bila, bilastine; fexo, fexofenadine; olop, olopatadine; oxat, oxatomide; levce, levocetirizine; bepo, bepotastine. Data are expressed as means ± SEM (*n* = 8 for control and olopatadine, *n* = 7 for bilastine, fexofenadine, levocetilizine, and bepotastine; *n* = 6 for oxatomide). **, *p* < 0.01 vs. control.

**Figure 6 pharmaceutics-12-00525-f006:**
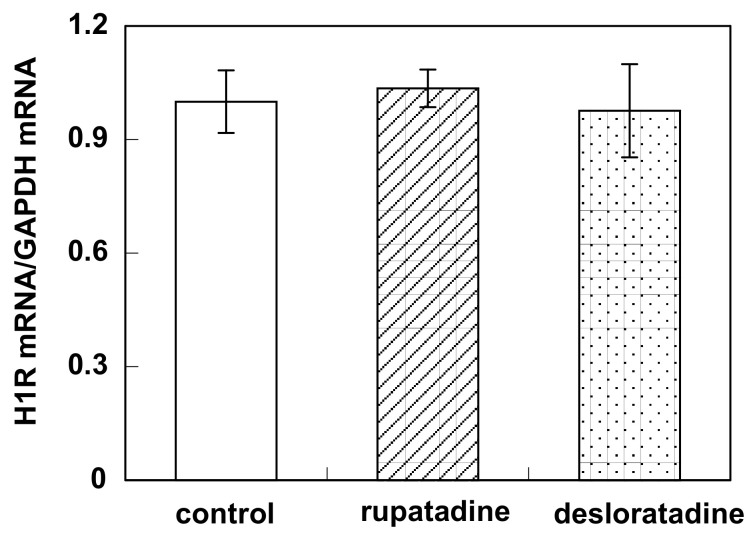
Effect of rupatadine and desloratadine on H1R gene expression in HeLa cells. HeLa cells were treated with or without 10 μM of rupatadine and desloratadine for 4 h. Then the cells were harvested and H1R mRNA was determined by real-time RT-PCR. Data are expressed as means ± SEM (*n* = 5).

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
