# Peer review of "Elucidation of Inverse Agonist Activity of Bilastine"

_pharmaceutics, 2020, doi:10.3390/pharmaceutics12060525_

Round 1

Reviewer 1 Report

The paper describe the pharmacology of bilastine demonstrating that this compound is an inverse agonist more than an antagonist. The study is well conduct, I just have some minor comments to improve the overall comprehensiveness of the paper and of the study design. First of all I would suggest the authors to explain the dose selection, even considering that just three doses have been used (the minimum doses to obtain a dose-response curve). Similarly, the comparison with other anti-H1 is very interesting, but it would be very informative if the author would justify the selection of these compounds over all the anti-H1. This point would be more specifically commented in the discussion section providing an hypothesis of the differences observed between the different compounds and how these differences could be translated into any clinical differences. 

Finally, in my opinion the abstract should be revisited, starting from the phrase " Antihistamines antagonize histamine and prevent it from binding to the histamine H1 receptor (H1R)". I think it would be better to explain it more precisely: an antihistamine could prevent the binding of histamine with one or all of the histamine receptors, not only the H1R. I understand that the authors are referring at antihistamine as the classical so call "antihistamine drugs" that means specifically acting on the H1R, but a more precise definition would help a reader independently from its knowledge about.

Author Response

(1) The paper describe the pharmacology of bilastine demonstrating that this compound is an inverse agonist more than an antagonist. The study is well conduct, I just have some minor comments to improve the overall comprehensiveness of the paper and of the study design. First of all I would suggest the authors to explain the dose selection, even considering that just three doses have been used (the minimum doses to obtain a dose-response curve). Similarly, the comparison with other anti-H1 is very interesting, but it would be very informative if the author would justify the selection of these compounds over all the anti-H1. This point would be more specifically commented in the discussion section providing an hypothesis of the differences observed between the different compounds and how these differences could be translated into any clinical differences.

Reply: Corcóstegui et al. reported the pharmacological characterization of bilastine (Corcóstegui et al., Drugs in R & D. 2005; 6: 371–384.), in which they showed that IC50 value of bilastine for [3H]mepyramine binding was 0.18 mM.  We selected the dose of bilastine with reference to that of their experiment using HEK293 cells over expressing human histamine H1 receptor (please refer to Figure 3 in their paper).  As mentioned in the text, previously, we showed that carebastine, mepyramine, fexofenadine, cetirizine, chlorpheniramine, diphenhydramine, and epinastine were inverse agonists and oxatomide, loratadine, and olopatadine were neutral antagonists (ref 9-11).  According our previous data, we selected fexofenadine and levocetirizine as controls of inverse agonists, and olopatadine and oxatomide as controls of neutral antagonists.  Rupatadine is partially metabolized to desloratadine that is an active form of loratadine.  Bepotastine is one of the most prescribed H1-antihistamines in Japan.  This is the reason why we selected these H1-antihistamines.  We have described this on page 7, lines 308-314.

(2) Finally, in my opinion the abstract should be revisited, starting from the phrase " Antihistamines antagonize histamine and prevent it from binding to the histamine H1 receptor (H1R)". I think it would be better to explain it more precisely: an antihistamine could prevent the binding of histamine with one or all of the histamine receptors, not only the H1R. I understand that the authors are referring at antihistamine as the classical so call "antihistamine drugs" that means specifically acting on the H1R, but a more precise definition would help a reader independently from its knowledge about.   

Reply: As reviewer suggested, we have changed the word “antihistamines” to “H1-antihistamines” to show that these drugs specifically bind to H1R.

Reviewer 2 Report

In this study, the authors aimed to elucidate the inverse agonistic activity of bilastine. The authors have HeLa cells, which is an appropriate model to study the intended outcomes.

  1. The authors write in the abstract (background) “Here we evaluated whether bilastine has inverse agonistic activity or not “. Indeed, it has already been reported previously that bilastine is an “inverse agonist” and authors also mention this in the results 3.1 “Pretreatment with bilastine shown as inverse agonist previously (10)”. Therefore, the author’s claim that they aimed to investigate whether Bilastine has any inverse agonist, is not correct. In contrast, the effect of Bilastine on inositol phosphatase formation is new, and perhaps this can be the main question when elucidating inverse agonistic activity.
  2. The abstract should be modified stating unanswered core question such as “there is no information previously available if bilastine has any effects on inositol phosphatase formation”
  3. The figures are not clear, very low resolution and font size is very small making it impossible to understand. Please increase the font size, resolution including lines in Figure 1A, 2A, and 3A.

Results

Results 3.1 :

  1. Results are not properly formulated, there are few grammatical mistakes. For instance, the first sentence for Fig. 1A makes no sense, sentence is not complete.
  2. Authors state “Stimulation of cells-----increase in the“, should be written as “Stimulation of cells ----increased
  3. Authors state “Bilastine significantly decerase---“, should be written as ““Bilastine significantly decerased”
  4. There is some redundant/confusing text stating “Schemes follow the same formatting. If there are multiple panel ------ “ line 164-167 page 4.

Results 3.2:

  1. Increase the resolution of the fonts in Figure 4AB.
  2. Figure 4A-B are wrongly presented, the Y-axis should be numbered to 0-100 as it the data presented is % of control. If Y-axis is expressed in ODs, it should be mentioned in the Y-axis label as “IP formation (OD)”

Discussion

  1. Overall, discussion requires improvement, not to confuse the statements with your results and previously reported data.
  2. For example, line 311, page 7: Sentence starting “Using HeLa cell system, we elucidated that------(9-11)” should be modified as “ Using HeLa cell system, we have previously reported/elucidated-----(9-11)”.

Author Response

(1) The authors write in the abstract (background) “Here we evaluated whether bilastine has inverse agonistic activity or not “. Indeed, it has already been reported previously that bilastine is an “inverse agonist” and authors also mention this in the results 3.1 “Pretreatment with bilastine shown as inverse agonist previously (10)”. Therefore, the author’s claim that they aimed to investigate whether Bilastine has any inverse agonist, is not correct. In contrast, the effect of Bilastine on inositol phosphatase formation is new, and perhaps this can be the main question when elucidating inverse agonistic activity.

Reply: Firstly, we apologize because of our poor English.  Our poor English confused you.  In the reference #10, we reported that fexofenadine is an inverse agonist.  There is no data for bilastine in reference #10 paper.  Thus, in the present study, it is the first time to show that bilastine is an inverse agonist. We have changed the sentence as follows; “Pretreatment with bilastine or fexofenadine attenuated fluorescence oscillation observed in the cells at rest (Fig. 2A).”

(2) The abstract should be modified stating unanswered core question such as “there is no information previously available if bilastine has any effects on inositol phosphatase formation”

Reply: As reviewer suggested, we have changed this sentence as follows (p1, lines 20-21); “However, there is no information to show the inverse agonistic activity of bilastine including inositol phosphates accumulation, and its inverse agonist activity is yet to be elucidated.

(3) The figures are not clear, very low resolution and font size is very small making it impossible to understand. Please increase the font size, resolution including lines in Figure 1A, 2A, and 3A.

Reply: We have revised all Figures in this manuscript.

Results

Results 3.1 :

(4) Results are not properly formulated, there are few grammatical mistakes. For instance, the first sentence for Fig. 1A makes no sense, sentence is not complete.

Reply: We have changed this sentence as follows; “Intracellular Ca2+ oscillation was observed in HeLa cells at rest

(5) Authors state “Stimulation of cells-----increase in the“, should be written as “Stimulation of cells ----increased

(6) Authors state “Bilastine significantly decerase---“, should be written as ““Bilastine significantly decreased”

Reply to comments #5 and #6: We have changed the words “increase” and “decrease” to “increased” and “decreased”.  In addition, we have changed the word “tend” to “tended” (p4, line 191 and p6, line 264).

(7) There is some redundant/confusing text stating “Schemes follow the same formatting. If there are multiple panel ------ “line 164-167 page 4.

Reply: We have deleted these sentences because they are part of the instruction for authors, and we failed to delete them.

Results 3.2:

(8) Increase the resolution of the fonts in Figure 4AB.

Reply: We have revised all Figures in this manuscript.

(9) Figure 4A-B are wrongly presented, the Y-axis should be numbered to 0-100 as it the data presented is % of control. If Y-axis is expressed in ODs, it should be mentioned in the Y-axis label as “IP formation (OD)”    

Reply: We agree reviewer’ comment.  This should be “fold of control”.  We have revised this.

Discussion

(10) Overall, discussion requires improvement, not to confuse the statements with your results and previously reported data.  For example, line 311, page 7: Sentence starting “Using HeLa cell system, we elucidated that- -----(9-11)” should be modified as “Using HeLa cell system, we have previously reported/elucidated-----(9-11)”.

Reply: This comment is due to misunderstanding that comes from our poor English.  As we described above, it is the first time to show that bilastine is an inverse agonist.  So, we do not think to revise discussion.  However, as reviewer suggested, we have changed this sentence as follows (p7, lines 308-314); “Using HeLa cell system, we have reported that carebastine, mepyramine, fexofenadine, cetirizine, chlorpheniramine, diphenhydramine, and epinastine were inverse agonists and oxatomide, loratadine, and olopatadine were neutral antagonists [9-11].

Reviewer 3 Report

Dear Authors

The manuscript by Mizuguchi et al. is an interesting study on the investigation of inverse agonist effect of bilastine as H1R binder.

The topic is noteworth and available for publication in this journals because it falls in the aim and scope.

Main text has been organized according to journal guidelines and fluent in content. However figures resolution should be improved, figure 5 and 4 are hard to read.

Also be advise that bilastine structure should be added and a little focus on SAR properties (Stefanucci et al. Int J Mol Sci. 2011) should be done. Consider also to discuss in general compounds with H1R affinity, such as trypthophane derivatives. 

 Overall my recomendation is minor revision.

Author Response

(1) However, figures resolution should be improved, figure 5 and 4 are hard to read.

Reply: We have revised all Figures in this manuscript.

(2) Also be advise that bilastine structure should be added and a little focus on SAR properties (Stefanucci et al. Int J Mol Sci. 2011) should be done. Consider also to discuss in general compounds with H1R affinity, such as trypthophane derivatives.

Reply: We have added the structure of bilastine as Fig. 1A.  We have added the following sentences in Discussion (p7, lines 343 -p8, line 357); “Although structure-activity relationships (SAR) of H1-antihistamines have been well investigated, little information is available concerning SAR for their inverse agonistic activity.  There are 6 main chemical groups of H1-antihistamines: alkylamines, ethanolamines, ethylenediamines piperazines, piperidines, and phenothiazines (Yanai K, Yoshikawa T, Yanai A, Nakamura T, Iida T, Leurs R, et al. The clinical pharmacology of non-sedating antihistamines. Pharmacol Ther 2017; 178:148-56.).  Bilastine belongs to piperidine derivatives although it is not structurally derived from any other currently available antihistamines (Ridolo E, Montagni M, Bonzano L, Incorvaia C, Canonica GW. Bilastine: new insight into antihistamine treatment. Clin Mol Allergy 2015; 13: 1 doi: 10.1186/s12948-015-0008-x.).  Fexofenadine and bepotastine are also piperidine derivatives, thus, piperidine moiety may contribute inverse agonist activities.  However, desloratadine, another piperidine derivative, did not show inverse activity, suggesting the necessity of additional moiety for inverse agonist activity.  We showed that levocetirizine belongs to piperazine is an inverse agonist, while, oxatomide belongs to same group as levocetirizine is neutral antagonist.  In addition, we reported that mepyramine, diphenhydramine, and chlorpheniramine (belongs to ethylamines, ethanolamines, and alkylamines, respectively) were also inverse agonists (9-11).  Thus, it is difficult to explain which structures are contribute to inverse agonist activities of H1-antihistamines.  Further studies should be addressed to SAR of inverse agonist activity of H1-antihistamines to understand the molecular mechanism how inverse agonists stabilize the inactive form of H1R.  It was reported that some indolylpiperidines show H1-antagonistic activity (Fonquerna S, Miralpeix M, Page`s L, Puig C, Cardu ́s A, Anto ́n F. Synthesis and Structure-Activity Relationships of Novel Histamine H1 Antagonists: Indolylpiperidinyl Benzoic Acid Derivatives. J. Med. Chem. 2004; 47: 6326-6337.  Examination of their inverse agonist activity may provide some useful insights.”

Round 2

Reviewer 2 Report

The authors have satisfactorily responded to all comments but still, at few places, there are grammatical mistakes which should be corrected.  For example:

Line # 198: "suppressed the increase in IPs formation", should be "suppressed IPs formation "

Line # 200: "as compared with control"  should be "as compared to control"

Fig 1A and 3A, on Y-axis instead of " fluorescence " alone, I believe it should be labelled as " fluorescence (intracellular Ca2+)"

Author Response

The authors have satisfactorily responded to all comments but still, at few places, there are grammatical mistakes which should be corrected.  For example:
Line # 198: "suppressed the increase in IPs formation", should be "suppressed IPs formation "

Line # 200: "as compared with control" should be "as compared to control"

Fig 1A and 3A, on Y-axis instead of " fluorescence " alone, I believe it should be labelled as " fluorescence (intracellular Ca2+)"

We have revised Lines #198 and #200. 

Also, we have revised some mistakes (p7, line 324, “a neutral”; p8, 352, “structures contribute”.

We have changed the label of the Y-axis in Fig. 1~3 to “fluorescence (intracellular Ca2+)”.

We have changed the font style of titles of figures from regular to bold.